# Test-System for Bacteria Sensing Based on Peroxidase-Like Activity of Inkjet-Printed Magnetite Nanoparticles

**DOI:** 10.3390/nano10020313

**Published:** 2020-02-12

**Authors:** Maxim Zakharzhevskii, Andrey S. Drozdov, Denis S. Kolchanov, Liubov Shkodenko, Vladimir V. Vinogradov

**Affiliations:** Laboratory of Solution Chemistry of Advanced Materials and Technologies, ITMO University, 197101 St. Petersburg, Russia

**Keywords:** magnetite, nanoparticles, enzyme-like activity, bacteria detection, inkjet

## Abstract

Rapid detection of bacterial contamination is an essential task in numerous medical and technical processes and one of the most rapidly developing areas of nano-based analytics. Here, we present a simple-to-use and special-equipment-free test-system for bacteria detection based on magnetite nanoparticle arrays. The system is based on peroxide oxidation of chromogenic substrate catalyzed by magnetite nanoparticles, and the process undergoes computer-aided visual analysis. The nanoparticles used had a pristine surface free of adsorbed molecules and demonstrated high catalytic activities up to 6585 U/mg. The catalytic process showed the Michaelis–Menten kinetic with *K_m_* valued 1.22 mmol/L and *V_max_* of 4.39 µmol/s. The nanoparticles synthesized were used for the creation of inkjet printing inks and the design of sensor arrays by soft lithography. The printed sensors require no special equipment for data reading and showed a linear response for the detection of model bacteria in the range of 10^4^–10^8^ colony-forming units (CFU) per milliliter with the detection limit of 3.2 × 10^3^ CFU/mL.

## 1. Introduction

Pathogenic bacteria cause many globally essential diseases. Even if the number of dangerous pathogens is negligible in a sample, it can lead to severe consequences [1]. Rapid detection of hazardous bacteria in air, water, and food can improve the quality of clinical diagnosis and reduce the rate of disease transmittance. Another important issue is a high cost of bacteria tests, as the majority of bacterial diseases are found in the third world countries [2,3]. Unfortunately, modern methods for bacteria detection feature several disadvantages, such as high cost, long-time procedures, etc. A standard technique for bacteria determination is based on culturing and plating, which is relatively inexpensive, but it requires lengthy incubation for reliable results [4]. In its turn, reducing detection time and increasing overall sensitivity may be developed by the application of molecularly-amplified (eq polymerase chain reaction) and enzyme-amplified (eq enzyme-linked immunosorbent assay) methods. However, these techniques are expensive and require specialized laboratory facilities [5]. Currently, big efforts are focused on the application of nanomaterials and their properties to overcome the drawbacks other methods have [6]. There are two main strategies. The first class of methods is based on the interactions of bacterial metabolites or bacterial cell walls and nanomaterial. Such an interaction changes physicochemical properties of the latter [7,8]. For instance, Ag and Au nanoparticles can be used for surface-enhanced Raman scattering (SERS) [9], carbon nanotubes for potentiometry [10], CdSe [11], and CdS [12] quantum dots in the fluorescence microscopy and other materials and methods have found their application in bacteria sensing as well [13,14].

The second approach focuses on the optimization of standard biomedical methods using nanomaterials as artificial biomolecular analogues. The methods, which use enzyme mimetics combining the high activity of natural enzymes with comparative cheapness and stability of inorganic catalysts, are actively explored [15]. The pioneering work of Gao et al., has shown that the ability of magnetite particles to catalyze the decomposition of hydrogen peroxide is close to the enzymatic activity of horseradish peroxidase (HRP) [16]. Developing of the area has led to the discovery of other inorganic enzyme mimics with catalase-like, oxidase-like, and phosphatase-like activities [17,18,19]. Such materials and composites find their application in designing biosensors towards various analytes ranging from small molecules to the whole cells with the working principle based on promotion or inhibition of catalytic activities [20,21,22]. This principle can also be used for bacteria detection, and modern trends are aimed at increasing the effectiveness and sensitivity of detection systems and at simplifying analysis processes, thus moving to the point of care diagnostics [23,24,25].

In this article, we describe a simple test-system for bacteria detection which can be used without sophisticated analytical equipment (Figure 1). The system described is based on a stable magnetite hydrosol with a pristine surface and extremely high peroxidase-like activity, which was patterned into sensor arrays by inkjet printing on polyethylene terephthalate (PET) substrates. The bacteria detection used the inhibition of the sensor peroxidase activity and showed linear dependence in the range of 10^4^–10^8^ colony-forming units (CFU) per milliliter with the detection limit of 3.2 × 10^3^ CFU/mL. Physico-chemical characteristics of the nanoparticles and enzyme-like activity were measured and discussed.

## 2. Materials and Methods

### 2.1. Chemicals

Iron(II) chloride tetrahydrate >98.5% iron (III) chloride hexahydrate >99%, and 2,2′-azino-bis(3-ethylbenzothiazoline-6-sulphonic acid) (ABTS, >99%) were obtained from Sigma-Aldrich and used without further purification. Diethylene glycol (DEG), glycine hydrochloride, aqueous ammonia solution >27.5%, and hydrogen peroxide >35% were purchased from Vekton (Saint-Petersburg, Russia).

### 2.2. Synthesis of Magnetite Nanoparticles (MNPs)

The Fe_3_O_4_ nanoparticles were synthesized by a coprecipitation method, as previously described [26,27]. An amount of 2.5 g FeCl_2_·4H_2_O and 5 g of FeCl_3_·6H_2_O were dissolved with 100 mL of deionized water. Then, 11 mL of aqueous ammonia solution was added to the solution under constant stirring (500 rpm) at room temperature and mixed for 1 min. The magnetite nanoparticles were collected using a magnet and washed using deionized water until a neutral pH level. Magnetic separation included repeated stages: Particle deposition by applying a magnetic field, decantation, adding water, and mechanical resuspension of the sediment. The resulting black precipitate was mixed with 100 mL of deionized water and subjected to ultrasonic treatment (30 kHz, 110 W) under constant stirring for 2 h. Finally, the solution of magnetite nanoparticles was cooled down to room temperature. The ensuing solution of magnetite nanoparticles exhibited the magnetic liquid properties, thus, when a magnetic field was applied, the particles did not separate from the water. The mass fraction of magnetite in the resulting colloid solution was 2% wt.

### 2.3. Inkjet Printing of MNPs

To pattern MNPs by inkjet printing, the inks with proper rheological parameters were prepared in the manner described earlier [28]. For that purpose, MNPs hydrosol was mixed with diethylene glycol in a ratio of 1:1 to increase its viscosity and decrease the surface tension of the ink to form a steady drop. The hydrodynamic radius of magnetite particles in ink did not exceed 25 nm. The solution was thoroughly mixed and cleaned from possible impurities using a filter with a pore size of 0.22 microns before loading into the cartridge. The cartridge was degassed in the desiccator under vacuum to avoid the formation of residual air bubbles during printing. The test system for bacteria sensing was printed on a Dimatix 2831 R&D inkjet printer, using Dimatix Material cartridges 11610 with a drop volume of 10 pL. The frequency of droplet formation was set to 20 kHz, and the nozzle voltage was set to 19 mV, to achieve a drop rate of 5 m/s.

### 2.4. Hydrodynamic Size and Zeta Potential Measurements

Hydrodynamic radii and zeta potentials were measured by dynamic light scattering and electrophoretic light scattering techniques, respectively, using a Photocor Compact-Z analyzer (Moscow, Russia). All experiments were conducted under the thermostatic conditions at temperature 25 °C, laser beam power 15 mW, and the time of measurements of 40 s. A 90-degree scattering angle was used to determine the particle size, and a 20-degree angle was used to estimate the zeta potential. Before measurements, equal amounts of hydrosol or xerogel MNPs were diluted in Milli-Q water until transparent solutions were obtained.

### 2.5. Evaluation of MNPs Colloidal Stability at Different pH Levels

To evaluate the colloidal stability of MNPs in various media, the hydrosol of MNPs was diluted with solutions of sodium hydroxide or hydrochloric acids until the desired pH level was achieved. The solutions were analyzed using dynamic and electrophoretic scattering methods to obtain hydrodynamic characteristics. The experiments were conducted in the manner described in Section 2.4.

### 2.6. Enzymatic Activity of MNPs

The peroxidase-like activity of freshly synthesized magnetite nanoparticles was determined spectrophotometrically by catalytic oxidation of chromogenic substrate ABTS in the presence of H_2_O_2_ at room temperature. The as-prepared hydrosol was dissolved in the glycine buffer (pH = 3.5) to a final concentration of 40 µg/mL. To 1.8 mL of MNPs hydrosol, 100 µL of ABTS (25 mg/mL) and 100 µL of H_2_O_2_ (40 mM) solutions were added, the optical density of the system was measured at 415 nm in a kinetic mode. Control experiments included the examination of optical densities of the mixtures of ABTS and MNPs, MNPs and H_2_O_2_ (2.0 mM), and ABTS (1.2 mM) with H_2_O_2_ (2.0 mM) in 20 mM glycine buffer (pH = 3.5), respectively. To compare the activities of different transition states of MNPs as-prepared hydrosol, xerogel, inks, and dried inks were tested in the same conditions.

### 2.7. Steady-State Kinetics Assay

Steady-state kinetic measurements were measured in a kinetic mode at 415 nm under the standard reaction conditions (20 mM Gly buffer, pH = 3.5, 25 °C, 40 µg/mL Fe_3_O_4_ hydrosol) by varying the concentration of ABTS at a fixed concentration of H_2_O_2_ or vice versa. The Michaelis–Menten constant and kinetic parameters were calculated using Lineweaver–Burk linearization plots.

### 2.8. Colorimetric Detection of Bacteria

Colorimetric detection of bacteria was established based on the inhibition of MNPs peroxidase mimics. For the experiment, the bacteria were preliminarily separated from the medium by centrifuging and replacing the medium with a buffer solution, after which sequential dilution was performed to obtain samples with different concentrations of bacteria. Typically, 10 µL of MNPs with a concentration of 40 µg/mL and 10 µL of aliquot with different concentrations of *E. coli* were added into a 2 mL solution of 1.2 mM ABTS and 20 mM glycine buffer (pH = 3.5). The maximum value of optical density (concentration of oxidized ABTS) of the mixture obtained was plotted against the bacteria concentration. The control CFU measurements were done by culturing and plating.

### 2.9. Image-Based Measurements of Printed Sensors

An amount of 35 µL ABTS-H_2_O_2_ mixture in a glycine buffer (pH = 3.5) was dropped using a multichannel pipette onto the printed sensor and was kept at 25 °C for 10 min on a backlighting table. The standard mixture includes 10 mM H_2_O_2_ and varied concentrations of ABTS acting as chromogenic substrate. Images were taken with a digital camera in a manual mode and were converted into the grayscale mode in Photoshop. The luminance of the samples was measured with the Color Sampler Tool in the center of drops with the sample size of 31 × 31 pixels.

### 2.10. Image-Based Bacteria Detection

Following the calibration chart, *E. coli* concentration was varied. All measurements were conducted under standard conditions, including 20 mM glycine buffer, pH = 3.5, 25 °C, and 10 mM of hydrogen peroxide, 1.2 mM of ABTS by varying the concentration of *E. coli* from 10^4^ to 10^8^ CFU/mL. The samples were incubated for 10 min and analyzed.

### 2.11. Characterization

Specific surface area, average pore diameter, and total pore volume were determined by the nitrogen adsorption–desorption method using Quantachrome Nova 1200(e). Sample preparation included samples degassing for 2 h at 100 °C. The surface area was calculated using the Brunauer–Emmett–Teller (BET) equation; the average pore diameter was calculated by the method of Barrett–Joyner–Halenda (BJH). For the comparison, the xerogel was obtained by evaporation at 70 °C in an air oven at normal pressure. The crystal phase was characterized by an X-ray diffractometer D8 Advance of Bruker AXS using Cu *Ka* radiation (λ = 1.54 Å). The specimen was characterized by TEM using a high-resolution TEM Tecnai F20 G2 and by SEM using (HR-SEM) Sirion. Spectrophotometric evaluations were made by Agilent Cary 8454 UV–Visible spectrophotometer with a thermostatic cuvette chamber. Digital images were taken with a Nikon D5300 camera, AF-S DX Nikkor 18–55 mm f/3.5–5.6G VRII shot at 31 mm, manual exposure, 1/100 s, f/4.5, ISO 400, compensation: −1/3.

## 3. Results and Discussion

### 3.1. Physical Properties of MNPs

The stable hydrosol of magnetite nanoparticles was obtained by the US-assisted coprecipitation procedure using a mixture of ferric and ferrous chlorides and water–ammonia solution as a base [29]. The precipitate was washed with deionized water by magnetic separation to eliminate nonmagnetic by-products and excess of ammonia. The resulting material was dispersed in 15 MOhm deionized water and subjected to ultrasonic treatment for 2 h (see Section 2.2 for details). Ultrasonication of the particles in water led to the disintegration of the aggregates formed during the coprecipitation process, and intense hydroxylation of the particle surface. As a result, a black stable colloidal system with a magnetic fluid-like behavior was formed. Dynamic light scattering showed that the solution consisted of monodisperse nanoparticles with a hydrodynamic radius of 35 ± 3 nm. The kinetic stability of the resulting colloidal solution was confirmed by electrophoretic light scattering, which shows that zeta potential was 36 ± 2 mV.

To further study the composition and physical properties, the colloidal solution was dried in a vacuum for several hours to avoid oxidation. The thermogravimetric study revealed that the mass fraction of magnetite nanoparticles in the solution was >2%. The X-ray diffraction pattern of the material corresponded to the crystal phase of magnetite, peaks referred to JCPDS file No. 19-0629 (Figure 2A). According to the Scherrer equation, the crystallite size was 10.9 nm. The analysis of particles via TEM and SEM proved a narrow particle size distribution with the diameter of crystallites 10 ± 3 nm. (Figure 2B,C). Upon solvent removal, MNPs underwent sol-gel transition and formed a mesoporous xerogel matrix with the total surface area of 120 m^2^/g and a mean pore diameter of 8 nm (ESI Appendix A). Such xerogels were stable in water solutions without any signs of resuspension. The magnetization curve of the material showed superparamagnetic behavior with magnetization up to 78 emu/g at 8 kOe (Figure 2D).

Excellent colloidal stability and absence of surface modification with organic molecules determined the prospects of the synthesized MNPs towards hydrogen peroxide catalytic cleavage. The colorimetric reaction of ABTS oxidation by hydroxyl radicals resulting in the emerald-green coloring of the solution with the maximum absorbance at 415 nm was used to explore the peroxidase-like activity of the material [30,31]. The initial examinations aimed to find optimal concentrations of H_2_O_2_ related to reaction rates. The reaction rate showed Michaelis–Menten dependence on hydrogen peroxide (Figure 3A,B) with *K_m_* measured for the system valued at 152 mmol/L. The dependence of the ABTS oxidation rate on its concentration was performed at the concentration of hydrogen peroxide of 10 mM to overcome the problems connected with over-oxidation of the substrate [32].

ABTS concentration was varied to investigate kinetic parameters of the process using the Michaelis–Menten model and Lineweaver–Burk linearization graphs [33] (See Section 2 for details). The dependence of the initial inverse velocity on the inverse concentration of a chromogenic substrate showed linear dependence for nanozyme; therefore, kinetics were described using the Michaelis–Menten model with calculated *K_m_* 1.22 mmol/L, *V_max_* 4.39 µmol/s, and *K_cat_* 365.8 s^−1^) (Figure 3C,D). The calculated activity of the magnetite hydrosol obtained was 6585 U/mg of dry magnetite, which exceeded the activity of the majority of commercially available enzymes [34]. Compared to the data from the literature for this substrate, the tested pristine nanoparticles showed higher reaction rates, which can be attributed to the higher activity of the surface due to the availability of catalytic centers and lower diffusion limitations (See ESI Appendix A).

### 3.2. Bacteria Detection

The principal for bacteria detection in this study relied on bacteria-induced inhibition of MNPs peroxidase activity. Therefore, adding different concentrations of washed bacteria to the solution modified the procedure of the catalytic experiment. The experiments showed that increasing the bacterial concentration decreases the rate of reaction, which reduces the slope of the kinetic curve. Bacteria inhibit the activity of magnetite nanocatalysts in the oxidation reaction of a chromogenic substrate ABTS-hydrogen peroxide. In this case, one can use the optical density at a wavelength of an analytical signal at a specific time. In this study, we chose a 4-min point that gave the most reproducible results (Figure 4).

The dependence of optical density plotted on the logarithm of the bacteria concentration was almost linear. Some deviations from the relationship were associated with increased turbidity in samples with a high level of bacteria. This method allowed determining the concentration of bacteria spectrophotometrically in the concentration range from 10^7^ to 10^3^ CFU/mL. To address the observed effect, evaluations were made to determine the influence of bacteria concentration on hydrodynamic parameters of MNPs. It was revealed that the interaction of positively charged MNPs with negatively (−34 ± 3 mV) charged bacteria decreased zeta potential from 36 ± 2 to −10 ± 2 mV (Figure 4B) and resulted in coagulation of the system. The mechanism of inhibition seems to be connected with the electrostatic interaction of positively charged particles of magnetite and the negatively charged surface of *E. coli* or/and waste products of bacteria. The same behavior was observed for pure MNPs with the pH level decrease (Appendix A). The isoelectric point of MNPs was supposedly defined as 8.3. Moreover, at pH 9, the hydrodynamic radius doubled, and then an almost exponential increase in the radius was observed, which corresponds to a faster aggregation rate. These results correlate with the literature data on decreasing of enzyme-like activity of magnetite nanoparticles in basic conditions [16]. 

### 3.3. Inkjet Printing of Bacteria Sensors

After preliminary evaluations with MNPs hydrosol, the efforts were made to prepare a model test system. The sensor was designed in the form of a ring with 7 mm in diameter; the line thickness was 1 mm. This design provides maximum convenience and practicality of the test system. The sensor arrays were formed by a soft lithographic approach via inkjet printing. Stable droplet formation is of great importance when formulating inks, determining their applicability not only for research material printers but also for conventional inkjet printers [35]. In our work, printing was provided by the selected waveform, at a drop rate of 7 m/s. To obtain a smooth layer, the distance between the centers of the drops was 20 microns. Patterns were printed in seven layers; the temperature of the object table was set at 45 °C for rapid solvent removal. As a substrate, white paper with density of 80 g/m^3^ was used; the paper was laminated with a PET film before printing.

Earlier it was demonstrated that rheologic parameters of the magnetite hydrosol could be optimized by mixing with diethylene glycol [28]. Figure 5A represents the main rheological parameters of the ink. The inks based on magnetite hydrosol showed good print quality (Figure 5C,D). Continuous printing was possible for 5 h, after which the cartridge was refilled and reused. The measured peroxidase-like activity of the prepared inks was 30% lower compared to pristine MNPs (ESI Appendix A). The decrease observed in the activity may be attributed to the formation of the DEG layer on the surface of MNPs, which was proved by ATR-IR. Two peaks (2857 and 2925 cm^−1^) of symmetric and asymmetric CH_2_ stretching bands are observed in the FTIR spectra of dried inks (See Appendix A).

In contrast to hydrosols, sol-gel matrices of magnetite demonstrated significantly lower peroxidase activity. Measured catalytic activity for the xerogel valued only 5% of the initial activity of the hydrosol (See Appendix A). Several factors determined the lower activity of xerogels. Firstly, in contrast with hydrosols, hydrogen peroxide decomposition was performed in heterogeneous conditions, which brought diffusional limitations to the system. In addition, it is known from the literature that the catalytic activity of iron oxide nanoparticles is strongly connected with its charge or surface modification. It was shown that due to the presence of the two negatively charged sulfonic acid groups ABTS better interact with the positively charged nanoparticles [36]. Moreover, there is a correlation between the rise of zeta potential of nanoparticle and catalytic activity of MNPs. In the case of the xerogel, the positive charge reduced to +14 mV, which probably led to a decrease in affinity for 2,2′-azino-bis(3-ethylbenzothiazoline-6-sulphonic acid). At the same time, the presence of DEG had a negligible effect on the catalytic activity of dried inks compared to MNPs-derived xerogels, resulting in almost equal catalytic activities of dried MNPs hydrosols and dried inks.

The initial assessments were done to determine the response linearity of the sensor to various concentrations of the substrate. For this purpose, ABTS solutions with concentrations of 0.2–1.2 mM were dropped in the center of the sensor arrays, and the analytical signal was taken after 10 min of incubation. In this work, we decided to rely on the visual analysis of the images without using special analytical equipment, and analyze the samples using a conventional digital camera. The digital images were taken and processed in similar conditions using Photoshop image processing programs (see Section 2 ). The brightness of the images was flattened, the green channel was desaturated, and the average luminance was measured in the grayscale mode (Figure 6A). As seen, the decrease of the ABTS concentration led to the lower coloration of the drops on the sensor and lower luminance of the processed images. The analysis of images showed linear independence of the luminance on the concentration of ABTS in the range 0.2–1.2 mmol/L with the adjusted r-squared value of 0.986 (Figure 6B).

The performance of the platform was validated against *E. coli*. As seen in Figure 6C, samples containing bacteria left were clearly brighter than control samples. All those test papers were recorded and then analyzed by the method described to calculate the average gray values, which are presented in Figure 6D. The sensor demonstrated linear dependence of the luminance on the concentration of model *E. coli* bacteria in the concentration range of 10^4^–10^8^ CFU with the adjusted r-squared value 0.976. The limit of detection (LOD) was calculated using the formula: *LOD* = *3Sa/b*, where S is the value of the standard deviation of the response and b is the slope of the standard curve within the linear range. According to this formula, the LOD of the method proposed was 3.2 × 10^3^ CFU/mL.

## 4. Conclusions

Here, we have described the first example of a scalable inkjet-printed peroxide sensor based on magnetite and its application for the bacteria detection. After initial calibrations and verifications by the means of optical spectroscopy, the resulting sensors may be read without any specific equipment using a conventional CCD camera. The sensor was produced from stable magnetite hydrocolloids which were synthesized by the US-assisted coprecipitation procedure and consisted of 10 nm nanoparticles of magnetite. Pristinity of the nanoparticles surfaces determined the high catalytic activity of the system valued 6585 U/mg, the activity was described by the Michaelis–Menten model with calculated values of *K_m_* 1.22 mmol/L, *V_max_* 4.39 µmol/s, and *K_cat_* 365.8 s^−1^. The inhibition of exhibited peroxidase activity was accompanied by reducing zeta potential, which is likely to be connected with interaction between nanoparticles and the *E. coli* membrane. The sensors were produced by inkjet printing on PET films and demonstrated linear response to model *E. coli* bacteria in the range of 10^4^–10^8^ CFU with calculated LOD 3.2 × 10^3^ CFU. Further studies will be aimed at automatization of the analytic system and integration with neural networks to design optical test systems.

## Figures and Tables

**Figure 1 nanomaterials-10-00313-f001:**
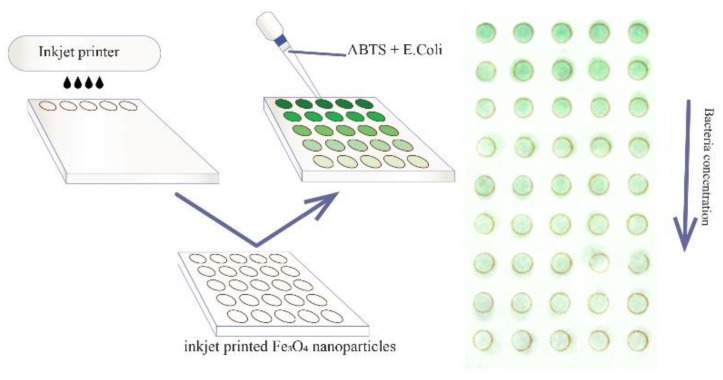
Principal scheme of the analytical system proposed.

**Figure 2 nanomaterials-10-00313-f002:**
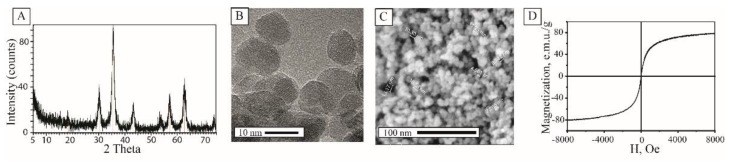
XRD pattern of the synthesized magnetite nanoparticles (MNPs) (**A**); TEM image of the particles (**B**); SEM image of the MNPs (**C**); magnetization curve of the material (**D**).

**Figure 3 nanomaterials-10-00313-f003:**
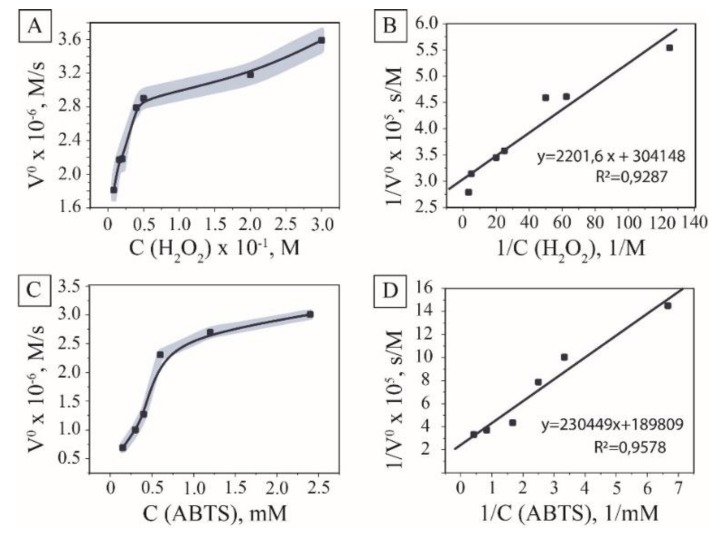
Peroxidase-like activity of MNPs. The kinetic curve and the double reciprocal plots of peroxidase activity at the constant concentration of H_2_O_2_ (**A**,**B**); the kinetic curve and the double reciprocal plots of peroxidase activity at the constant concentration of 2,2′-azino-bis(3-ethylbenzothiazoline-6-sulphonic acid) (ABTS) (**C**,**D**).

**Figure 4 nanomaterials-10-00313-f004:**
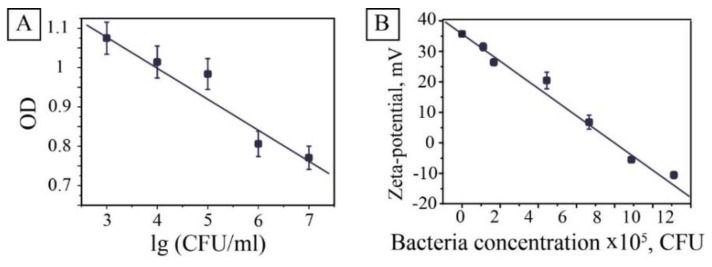
The effect of bacteria on peroxidase activity and colloidal stability of a solution of magnetite nanoparticles. Relationship of the optical density of the reaction mixture vs. decimal logarithm of the concentration of added *E. coli*. after 4 min of incubation (**A**); dependence of MNPs zeta potential against *E. coli* concentration (**B**).

**Figure 5 nanomaterials-10-00313-f005:**
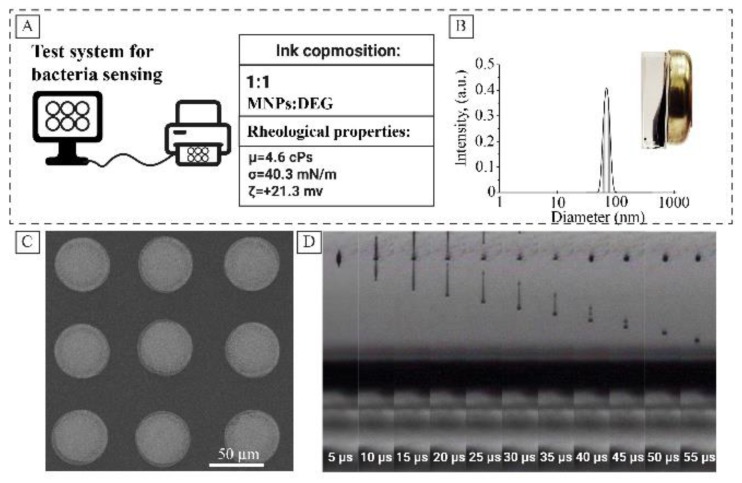
Inkjet printing ink based on magnetite hydrosol, basic rheological parameters (**A**); hydrodynamic radius of magnetite particles (**B**); SEM image of magnetite droplets on a roll substrate (**C**); life cycle of the droplet after leaving the nozzle (**D**).

**Figure 6 nanomaterials-10-00313-f006:**
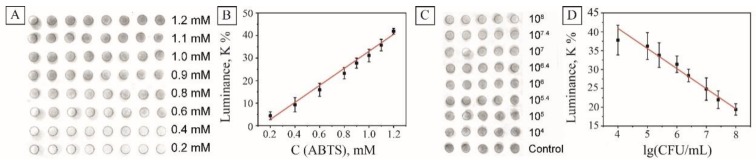
Device-free evaluation of the printed sensors. Processed image of the sensor array with various ABTS concentrations (**A**); the dependence of the measured sample luminance on ABTS concentration after 10 min of incubation (**B**); processed image of the sensor array in the presence of various amounts of *E. coli* CFU, (**C**); the dependence of the measured sample luminance on *E. coli* concentration after 10 min of incubation (**D**).

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
