# Peer review of "Test-System for Bacteria Sensing Based on Peroxidase-Like Activity of Inkjet-Printed Magnetite Nanoparticles"

_nanomaterials, 2020, doi:10.3390/nano10020313_

Round 1

Reviewer 1 Report

My suggestion is to accept after minor revision. Some specific comments are listed below –

In section 2.4, line 88-90, sentence needs to be rephrased. In section 2.9, line 120-121. There is no data included in the manuscript about surface area and BET, BJH. Please correct it. In section 3.1, line 147-146. It is not clear why the author suddenly compare the activity with xerogel. If it is for the explanation of the low activity of ink formulation, this paragraph should be placed in section 3.3. In section 3.1, line 183. How was the zeta potential of xerogel (+14 mV) determined? Should include in the method section. In section 3.2, line 191-192. There is redundant sentences found. Please proofread carefully. In section 3.3, line 210-212. No Figure 6 was found in the manuscript. Please correct it. In section 3.3, line 222. How was the activity of dry ink determined? Please address in the text or include in the method section. In section 3.3, line 231. Please include the original image (green dots) in the manuscript. It should be showing the capability of visual analysis for bacterial contamination. I would suggest showing in Figure 1 as part of schematic representation. It’s the most important and eye-catching data. Or, the author can include the raw image in the supplemental information.

Author Response

In section 2.4, line 88-90, sentence needs to be rephrased. In section 3.2, line 191-192. There is redundant sentences found. Please proofread carefully. In section 3.3, line 210-212. No Figure 6 was found in the manuscript. Please correct it.

The manuscript underwent additional rounds of proof-reading and was corrected accordingly.

In section 2.9, line 120-121. There is no data included in the manuscript about surface area and BET, BJH. Please correct it.

We are grateful for this comment, nitrogen adsorption isotherms were added to Supporting information figures S1 and S2.

In section 3.1, line 147-146. It is not clear why the author suddenly compare the activity with xerogel. If it is for the explanation of the low activity of ink formulation, this paragraph should be placed in section 3.3

We agree with the reviewer's opinion and move part of our text to another section according to this comment.

In section 3.3, line 231. Please include the original image (green dots) in the manuscript. It should be showing the capability of visual analysis for bacterial contamination. I would suggest showing in Figure 1 as part of schematic representation. It’s the most important and eye-catching data. Or, the author can include the raw image in the supplemental information.

We suggest that this comment can really improve the perception of our article so we decided to make new Figure 1 which includes raw data.

Reviewer 2 Report

The manuscript “Test-system for bacterial sensing based on peroxidase-like activity of inkjet-printed magnetite nanoparticles” by Zacharrzevskii et al., presents a magnetite nanoparticle array for bacterial detection. Bacterial cells are detected due to their negative effect on peroxide oxidation of chromogenic substrate catalyzed by magnetite nanoparticles. The reaction can be visualized by a computer-aided visual analysis.

This paper presents a potentially interesting work that quantitative and multiplex bacterial detection. I can recommend it to be published in Nanomaterials but the paper is missing control measurements.

The presence of bacterial cells has a negative effect on the catalyzed peroxide oxidation of chromogenic substrate by an unknown mechanism. Is this reaction specific? Is there any effect of fungal or plant cells on the reaction? Is the catalyzed peroxide oxidation also perturbed by some chemicals?

Other remarks:

DLS of MNPs was performed in pure water. However, bacterial detection is performed in a buffer solution. Hydrodynamic diameter of NPs depend on pH of the surrounding medium. For instance, NPs may aggregated at some pH, while being stable at others. I propose to perform DLS experiments at various pHs in order to correlate MNP stability with bacterial detection. Line 178-180, reference should be added for the relation between NP catalytic activity and their surface charge. Figure 4B. Bacterial surface charge is typically highly negative. The explication of the finding that bacterial zeta potential change from positive to negative with the increase of CFU/mL? If the measurements were done in the presence of MNPs this should be added in the legend. Also bacteria alone should be given. I steh modification of zeta potential a result of an additive effect: positive MNPs with negative bacterial cells? Check thought the text CFU or CFU/mL? (like line 240) E. Coli should be changed to E. coli everywhere in the text Line 203-243 Is it Fig. 6 or Fig. 5?

Author Response

The manuscript “Test-system for bacterial sensing based on peroxidase-like activity of inkjet-printed magnetite nanoparticles” by Zacharrzevskii et al., presents a magnetite nanoparticle array for bacterial detection. Bacterial cells are detected due to their negative effect on peroxide oxidation of chromogenic substrate catalyzed by magnetite nanoparticles. The reaction can be visualized by a computer-aided visual analysis.

This paper presents a potentially interesting work that quantitative and multiplex bacterial detection. I can recommend it to be published in Nanomaterials but the paper is missing control measurements.

Other remarks:

DLS of MNPs was performed in pure water. However, bacterial detection is performed in a buffer solution. Hydrodynamic diameter of NPs depend on pH of the surrounding medium. For instance, NPs may aggregated at some pH, while being stable at others. I propose to perform DLS experiments at various pHs in order to correlate MNP stability with bacterial detection.

We thank the reviewer for this comment. We make some additional experiments, prove aggregation processes and perform some extra DLS experiments at various pH (pages 3, 6 and Fig.S3, S4).

 Line 178-180, reference should be added for the relation between NP catalytic activity and their surface charge.

The Reference was added. We strongly recommend this article for a better understanding of our point of view doi: 10.1016/j.biomaterials.2009.05.005 and gave the corresponding reference in the text.

Figure 4B. Bacterial surface charge is typically highly negative. The explication of the finding that bacterial zeta potential change from positive to negative with the increase of CFU/mL? If the measurements were done in the presence of MNPs this should be added in the legend. Also bacteria alone should be given. I steh modification of zeta potential a result of an additive effect: positive MNPs with negative bacterial cells? Check thought the text CFU or CFU/mL? (like line 240) 

We are sorry for this incomprehensibility, we corrected the legend of our figures as well as gave an explanation in the text. Additional experimental experiments on the zeta potential of bacteria and MNPs were carried out and added to the article (page 6).

The presence of bacterial cells has a negative effect on the catalyzed peroxide oxidation of chromogenic substrate by an unknown mechanism. Is this reaction specific? Is there any effect of fungal or plant cells on the reaction? Is the catalyzed peroxide oxidation also perturbed by some chemicals?

As you correctly noted, the mechanism of this reaction is not defined. We suggest that this is simply an electrostatic interaction between positively charged magnetite and a negatively charged bacterial wall. Inhibition by other types of cells is assumed, but not yet studied since researchers are usually interested in the behavior of cells in the presence of nanoparticles, rather than nanoparticles in the presence of cells. The corresponding explanations were added to the article.

Reviewer 3 Report

The present manuscript has in attention a magnetite nanoparticle arrays for the detection of bacteria.

The authors are kindly requested to check the bibliographic references in the entire manuscript. There are some missing spaces between the words and the references (for example lines 25, 26, 29, 32, etc). Moreover, check if the bibliography contains all the necessary information (example, references 26 and 28 are missing the volume number, pages).

There are some abbreviations that should be defined in the manuscript (PTFE, CFU, MNP, etc.).

Under the sub-section 2.2, please give more details about the separation of the magnetite from the solution, and if there necessary additional treatments before inkjet printing.

In the case of the determined parameters, the authors are kindly requested to provide the standard deviation.

Please provide information about the stability of the xerogels in buffers.

In the Conclusions it is stated that „Here we described the first example of a scalable inkjet-printed peroxide sensor based on 246 magnetite and its application for detection of bacteria without any specific equipment.”  Please give more details, as in the manuscript are shown the Michaelis-Menten curves prepared based on the spectrophotometric measurements. 

Author Response

The authors are kindly requested to check the bibliographic references in the entire manuscript. There are some missing spaces between the words and the references (for example lines 25, 26, 29, 32, etc).

Typos in the text were corrected.

Moreover, check if the bibliography contains all the necessary information (example, references 26 and 28 are missing the volume number, pages).

The reference section was corrected

There are some abbreviations that should be defined in the manuscript (PTFE, CFU, MNP, etc.).

All was explained in the text.

Under the sub-section 2.2, please give more details about the separation of the magnetite from the solution, and if there necessary additional treatments before inkjet printing.

Additional details were described in the Materials and Methods section.

In the case of the determined parameters, the authors are kindly requested to provide the standard deviation.

Please provide information about the stability of the xerogels in buffers.

Thanks for this important issue! We conducted additional experiments on the stability of magnetic nanoparticles in various buffers with different pH (supporting information figure S3, S4). Redispersion of xerogel in water leads to the formation of an unstable solution with zeta potential +14.

In the Conclusions it is stated that „Here we described the first example of a scalable inkjet-printed peroxide sensor based on 246 magnetite and its application for detection of bacteria without any specific equipment.”  Please give more details, as in the manuscript are shown the Michaelis-Menten curves prepared based on the spectrophotometric measurements. 

Michaelis-Menten curves were unnecessary for bacteria detection, they were used for finding the best conditions for this reaction. To clarify this point the conclusion was corrected.     

Reviewer 4 Report

I cannot read this paper smoothly because of a lot of grammatical errors in English and careless mistakes, the use of unexplained abbreviatory words such as CFU, PTFE, BET, BJH, etc., and unkind Figure captions, although the subject of this paper seems interesting. The authors should have submitted this paper after thorough improvement and revision.

Other points:

Insert "Figure 5" into an appropriate position in the text, probably the paragraph starting from line 224.  The captions in particularly Figures 5 and 6 need more detailed explanation in connection with lines 224 to 243. This is the most imprtant part in this manuscript. I cannot understand what Figure 6D indicates. 

Author Response

I cannot read this paper smoothly because of a lot of grammatical errors in English and careless mistakes, the use of unexplained abbreviatory words such as CFU, PTFE, BET, BJH, etc., and unkind Figure captions, although the subject of this paper seems interesting. The authors should have submitted this paper after thorough improvement and revision.

The article was proofread and corrected accordingly.

Other points:

Insert "Figure 5" into an appropriate position in the text, probably the paragraph starting from line 224.  The captions in particularly Figures 5 and 6 need more detailed explanation in connection with lines 224 to 243. This is the most imprtant part in this manuscript. I cannot understand what Figure 6D indicates.  

We are grateful to the Reviewer for his comments. The article underwent another round of proof-reading

Round 2

Reviewer 2 Report

The authors reponded to all my comments.

Author Response

We are grateful to the reviewer for his comments

Reviewer 3 Report

The authors addressed to most of the reviewer's recommendation. 

Some of the references still needs to be completed (for example in the case of reference 26-Arabian Journal of Chemistry, Volume 13, Issue 1, 2020, Pages 1933-1944).

Author Response

We corrected all issues associated with reference section

Reviewer 4 Report

Please consider the following points.

line 29: What is 'e.q.' ?; line 32: application 'of'; line 36: '....cell walls the nonomaterias's.....' is not correct.; line 42: 'as artificial biomlecular analogues' ?; line 53: we 'describe'; line 168: seven times lower than 'that' by DLS ?; lines 95-96: 'by adding different concentrations of washed bacterial to the solution' ?; line 231: (Fig. 5 C,D)

Author Response

We are grateful for these comments. The article underwent another round of proofreading, grammar and style errors were corrected.